# A “Drug-Dependent” Immune System Can Compromise Protection against Infection: The Relationships between Psychostimulants and HIV

**DOI:** 10.3390/v13050722

**Published:** 2021-04-21

**Authors:** María Amparo Assis, Pedro Gabriel Carranza, Emilio Ambrosio

**Affiliations:** 1Facultad de Ciencias Médicas, Universidad Nacional de Santiago del Estero (UNSE), Santiago del Estero G4200, Argentina; pgcarranza@gmail.com; 2Laboratorio de Biología Molecular, Inmunología y Microbiología, Instituto Multidisciplinario de Salud, Tecnología y Desarrollo (IMSaTeD), CONICET-UNSE, Santiago del Estero G4206, Argentina; 3Departamento de Psicobiología, Facultad de Psicología, Universidad Nacional de Educación a Distancia (UNED), 28040 Madrid, Spain; eambrosio@psi.uned.es; 4Facultad de Agronomía y Agroindustrias, Universidad Nacional de Santiago del Estero, Santiago del Estero G4206, Argentina

**Keywords:** HIV, cocaine, amphetamines, dopamine, enkephalin, TLR4, T-cells, CD4^+^CD25^+^ T-cells, IL-17A

## Abstract

Psychostimulant use is a major comorbidity in people living with HIV, which was initially explained by them adopting risky behaviors that facilitate HIV transmission. However, the effects of drug use on the immune system might also influence this phenomenon. Psychostimulants act on peripheral immune cells even before they reach the central nervous system (CNS) and their effects on immunity are likely to influence HIV infection. Beyond their canonical activities, classic neurotransmitters and neuromodulators are expressed by peripheral immune cells (e.g., dopamine and enkephalins), which display immunomodulatory properties and could be influenced by psychostimulants. Immune receptors, like Toll-like receptors (TLRs) on microglia, are modulated by cocaine and amphetamine exposure. Since peripheral immunocytes also express TLRs, they may be similarly affected by psychostimulants. In this review, we will summarize how psychostimulants are currently thought to influence peripheral immunity, mainly focusing on catecholamines, enkephalins and TLR4, and shed light on how these drugs might affect HIV infection. We will try to shift from the classic CNS perspective and adopt a more holistic view, addressing the potential impact of psychostimulants on the peripheral immune system and how their systemic effects could influence HIV infection.

## 1. Introduction

Drug abuse and drug addiction are worldwide problems, often accompanied by devastating social, health and economic consequences. Over the past decades, most drugs of abuse (e.g., cocaine, amphetamines, opioids and cannabinoids) have been shown to alter certain functional aspects of the immune system, either directly or through neuro–immune mechanisms [1,2,3,4]. Research in this area was initially motivated by the observation that addicts and abusers are highly susceptible to viral, bacterial and fungal infections, and that they display numerous deficits in immune function, which can even increase their vulnerability to cancer [5,6,7]. Several reports have indicated that in experimental models, the immunosuppression induced by drugs of abuse may underlie the weaker host resistance to certain diseases. Some studies have even implicated drug abuse as a co-factor in susceptibility to infection by the human immunodeficiency virus (HIV) and hepatitis C virus [8,9,10,11,12]. Indeed, the relevance of the close relationship between events in the central nervous system (CNS) and the immune system, such as those triggered by drugs, becomes evident in these cases. In recent years, the effects of certain drugs of abuse and their endogenous counterparts in the brain has been described within the prism of neuroimmunology and immunopsychiatry.

### 1.1. Epidemiology of the Co-Occurrence of HIV Infection and Psychostimulant Use

Since the publication of the first cases of acquired immunodeficiency syndrome (AIDS) in the 1980s [13,14], more than 75 million people have been infected with HIV and there are currently around 40 million people living with HIV worldwide (according to the United Nations Program on HIV/AIDS—UNAIDS). From the outset, substance use disorder has been related to HIV transmission, as well as to poor engagement with care, minimal adherence to combination antiretroviral therapy (ART) and significant treatment failure [15,16,17]. According to the National Institute on Drug Abuse, people who come into contact with drugs or display high-risk behaviors associated with drug use are at a higher risk of acquiring HIV or hepatitis in different ways: (1) directly, by sharing contaminated needles or other nonsterile injection equipment; and (2) indirectly, when drugs impair judgment and lead to unprotected sex with an infected partner or with multiple partners.

The type of drug and the route of administration changes according to modes, cultures, economic income, religions, criminalization of consumption, street culture and racial and ethnic minorities. Among people who inject drugs, opioids and cocaine were strongly associated with HIV transmission in the 1980s, with different outbreaks over the past two decades [16,17,18]. The implementation of public health actions in many countries contributed to a 50% decline in the incidence of HIV in people who inject drugs, such as the syringe exchange programs and opioid substitution therapies [15,16]. As such, only 10% of HIV diagnoses currently occur in people who inject drugs [14].

By contrast, since the 1990s, there has been a worldwide increase in the relationship between noninjectable substances of abuse, like psychostimulants (cocaine, crack and amphetamine-like drugs), and HIV transmission. Concomitantly, such substance use has become much more frequent in the context of groups at a higher risk of acquiring HIV, such as: lesbian, gay, bisexual or transgender individuals; street youth; sex workers and low-income migrant workers [17,18]. Indeed, large studies indicated men who have sex with men that used substances, particularly stimulants, accounted for around 30% of new HIV infections [19,20]. In some regions, 15–50% of black and Latino men who have sex with men reported cocaine or methamphetamine use [21]. Furthermore, the strong association between alcohol use and HIV incidence cannot be overlooked, as well as the use of other substances like cannabinoids and that of new synthetic drugs that have no known impact on HIV transmission.

### 1.2. Neuroimmunology behind the Scenes

The CNS and the immune system carry out a wide variety of essential functions, often coordinated in the face of compromising situations, but that are necessary to preserve the homeostasis of the organism [22,23]. Both systems communicate through a complex network of chemical messengers capable of reaching independent anatomical locations, these include a broad repertoire of ligands (transmitters, modulators, cytokines) and receptors that facilitate communication both within and between these systems (Table 1) [24].

In 1984, Edwin Blalock proposed that the immune system should be considered as a sensory organ, a “sixth sense” capable of detecting agents with a sensitivity and specificity over and above that of the CNS [24,25,26]. The immune system can not only alert leukocytes to the presence of pathogens, tumor cells or allergens, but it can also inform the brain about events occurring in the periphery. The classical vision of the immune system as a defense shield has been extended to that of a system that contributes to the homeostasis of the organism, participating as a kind of “humoral branch” of the nervous system. The idea that the immune system does not function independently of the nervous and endocrine systems has become increasingly accepted, and the three coordinate a single integrated response to external and internal stimuli, whether physical or psychosocial [27,28].

Among others, immune cells can express: α- and β_2_-adrenoceptors [29]; nicotinic and muscarinic cholinergic receptors [30]; D1- and D2-type dopaminergic receptors (DARs) [31]; ionotropic and metabotropic glutamatergic receptors [32]; μ (MORs), δ (DORs) and κ opioid receptors [33,34]; and GABA_A_ receptors [35] (Table 1). In addition, tyrosine hydroxylase expression and the presence of monoamine transporters in the membrane of leukocytes has been described [36,37,38]. The activities mediated by all these proteins in immune cells are not yet fully understood, although it is possible that the biochemical and cellular events would be similar to those induced in the nervous system.

The modulation exerted on the immune system by the CNS has been explored extensively through numerous studies on the stress response, and it is currently known to be organized through neuronal and hormonal pathways [22,23,39]. The sympathetic nervous system (SNS) is the main neuronal pathway, with its noradrenergic fibers in close contact with lymphoid organs, while the hypothalamic–pituitary–adrenal (HPA) axis is the main hormonal pathway [40,41,42]. Catecholamines, neuropeptides and glucocorticoids are released peripherally by the activation of these pathways after central stimulation, and they are ultimately responsible for modulating the immune cells [43,44,45]. The effects provoked by these neurotransmitters and neurohormones (acting as immune transmitters/hormones) depend on the activation of the receptors that each immune cell can express [46,47].

Conversely, the influence the immune system exerts on the CNS is now beginning to be explored. The effect that T-cells and other immune cells can have on the brain and on behavior may originate in the periphery through soluble factors (cytokines, chemokines, hormones and transmitters) that cross the blood–brain barrier (BBB), but also, in situ by these cells transmigrating into the brain parenchyma. T-cells can modulate memory, plasticity, learning, neurogenesis and other CNS process [48,49,50], and there is some plasticity in the T-cell responses in the brain [51]. Recent findings also suggest meningeal adaptive immunity is critical for social behavior, spatial learning and memory [52]. Together, this evidence makes it necessary to reconsider the classic vision of the immune presence in the CNS, going beyond mere immune surveillance. Indeed, certain neuropathologies and neuropsychiatric disorders have been related to immune dysfunctions, and some of these disorders have been modulated by bone marrow transplantation [53,54,55,56,57], whereas in cases like depressive disorders an immunopathogenic cause has been proposed [58,59,60,61,62]. Moreover, peripheral immune markers capable of reflecting changes that also occur in the CNS may serve as valuable clinical tools to monitor brain pathologies peripherally. For example, the dopaminergic system expressed by peripheral blood lymphocytes has been studied in neuropsychiatric disorders (Parkinson’s disease, schizophrenia and alcoholism) to identify alterations in central dopamine (DA) transmission and monitor the effects of pharmacological treatments [63]. The alterations to the immune system associated with CNS disorders may play a role in the pathogenesis of these diseases, among which psychostimulant abuse and addiction can be included [4,64,65,66,67].

With regards to coincident HIV and psychostimulant use, and as mentioned before, people living with HIV who use drugs experience higher HIV-associated morbidity and mortality than non-users [68]. This increased vulnerability was often linked to the behavioral consequences of drug use, although new data indicates that the immunomodulatory effects of cocaine and amphetamines could also be involved in the progression of HIV infection [69,70,71,72]. There is considerable evidence supporting an influence of psychostimulants on immune cells and the cytokines they release, which may affect HIV pathogenesis, progression and mortality. However, classic neurotransmitters and neuromodulators may also be affected by psychostimulants, although they have only been studied in terms of their influence on the pathogenesis of HIV-associated neurocognitive disorders (HANDs) [68,73,74,75], ignoring that they are also expressed by peripheral immune cells. Other relevant molecules that have been studied at a central level are the toll-like receptors (TLRs), considered classical innate immune receptors that were recently implicated in certain drug effects on microglia [76,77]. On the other hand, HIV infection can aggravate the impact of the rewarding effects of psychostimulants since the HIV Tat protein produces a direct but reversible inhibition of DA transporter (DAT) activity in rat striatal synaptosomes [78,79]. Tat expression can provoke behavioral cross-sensitization to the locomotor effects of methamphetamine [80], and the long-term impact of Tat on the DA transmission and its drug-reinforcing effects may impair reward function, helping sustain the drug use/abuse that can lead to addiction [81]. However, all these findings were derived from the CNS and little is known about Tat’s effects at peripheral sites where DAT is expressed, such as immune cells.

Hence, this review will explore the potential modulation of peripheral neuro-immune transmitter signaling by psychostimulants, with a particular focus on catecholamines and enkephalins. In addition, since cocaine and amphetamine are thought to act on microglial TLR4, we will also contemplate the possible actions of psychostimulants via this receptor expressed peripherally. Thus, we will describe what is currently known about the immune functions modulated by catecholamines, enkephalins and TLR4, and how they might be affected by psychostimulants. Subsequently, we will explore them as elements that influence the noxious marriage between drug use and HIV infection. By considering the peripheral immunological targets of psychostimulants that also operate at the CNS level, we aim to offer an additional perspective that complements our current knowledge, potentially redirecting future therapies to treat HIV infection in drug users.

## 2. How Psychostimulants Can Influence Peripheral Immunity

### 2.1. Catecholamines

#### 2.1.1. Catecholamines as Neuro- and Immune-Transmitters

Dopamine, like other catecholamines, originates from the amino acid L-tyrosine. Tyrosine hydroxylase is the enzyme responsible for transforming L-tyrosine into L-dihydroxyphenylalanine (L-DOPA), this being the limiting step in the synthesis of catecholamines [82]. Rapidly, L-DOPA is transformed into DA that can generate norepinephrine if the cell expresses dopamine β-hydroxylase, which is the distinction between noradrenergic and dopaminergic cells. In neurons, the tyrosine hydroxylase activity is regulated by the levels of catecholamines, the availability of co-factors and the activation of D2-type presynaptic DARs. Once synthesized, catecholamines are incorporated into vesicles through the vesicular monoamine transporter (VMAT) [83] and upon arrival of an action potential, Ca^2+^ entry promotes their release into the synaptic cleft. After this, different mechanisms clear the neurotransmitter from the synaptic cleft, with reuptake through transporter proteins such as the DAT in the presynaptic membrane [84] being the most efficient way to terminate neurotransmission. Following reuptake, the neurotransmitter can be restored in vesicles for reuse or it can be degraded by the enzyme monoamine oxidase (MAO).

Both norepinephrine and DA meet the criteria to be considered as transmitters in the immune system, since the immune cells that contain them are capable of producing and inactivating them. Moreover, these cells release these transmitters upon stimulation, and there are receptors on both the target and releasing cells that are sensitive to agonists and antagonists [85]. However, how immune cells regulate catecholamine synthesis is different to that observed in neural cells. Endogenous synthesis of catecholamines in immunocompetent cells was first discovered in 1990s [86,87,88], suggesting that these transmitters act as autocrine or paracrine mediators in immune cells, and that they mediate the communication between these cells and the nervous system. The synthesis of catecholamines depends on the stimulation of immune cells [89], and in fact, it depends on the expression of tyrosine hydroxylase that is low basally but is enhanced by stimulation [37,90,91]. Thus, T-cell stimulation with phytohemagglutinin or Concanavalin A (Con A) triggers an increase in catecholamine due to enhanced tyrosine hydroxylase expression and activity, in turn driven by PKC and an increase in Ca^2+^ [37,91]. Subsequently, PKC-modulated tyrosine hydroxylase expression is inhibited by D1 DARs [92]. Hence, while neurons release DA stored in vesicles as a consequence of action potentials, it is synthesized and released de novo by immune cells when activated. However, vesicular storage of catecholamines has only been described in CD4^+^CD25^+^ regulatory T-cells, which like neurons, constitutively express tyrosine hydroxylase and they contain substantial levels of DA, norepinephrine and epinephrine stored in reserpine-sensitive compartments.

The DA released in an autocrine manner by CD4^+^CD25^+^ T-cells suppresses IL-10 and TGF-β synthesis by acting on D1-type DARs [36]. It was also demonstrated that DA, acting via D1 DARs preferentially expressed by Tregs, reduces their suppressive activity, as well as their adhesive and migratory capacities [93]. These results suggest that DA acts via its D1-type DARs to “inhibit the inhibition” of the immune response. Indeed, D_3_ receptors appear to participate in the immunosuppressive effects of Tregs at the gut mucosal level by inhibiting IL-10 release [94], while D1 modulation of CD8^+^ Tregs has also been shown [95]. Dopaminergic modulation has also been described in resting and activated T-cells, with an increase in D_1_, D_4_ and D_5_ receptors after TCR signaling, facilitating T-cell depolarization. In addition, a selective D1-type receptor agonist reduced chemotactic migration and the secretion of TNF-α, INF-γ, IL-1β, IL-2, IL-4, IL-6, IL-8 and IL-10 [96].

Catecholamines can also reach leukocytes from sympathetic nerve endings or the plasma. Noradrenergic fibers are in close contact with thymus and bone marrow, as well as with lymph nodes, spleen and mucosa-associated lymphoid tissue [39,41,42,97]. This organization provides the anatomical basis for the sympathetic regulation of immune organs. In particular, the innervation of the spleen mainly originates in the superior mesenteric and celiac ganglia, and the fibers enter the organ by surrounding the splenic artery, traveling along the vasculature and continuing through the trabeculae to form the trabecular plexuses. Sympathetic fibers are present between cells in the T area, in the marginal areas where macrophages and B-cells reside, and in the marginal sinus that represents the site of entry of lymphocytes to the spleen [98,99]. In the thymus, the vast majority of nerves are located around the vasculature, bordering the thymic cortex [100]. Sympathoadrenergic modulation of hematopoiesis in the bone marrow has also been described. Indeed, SNS nerves release noradrenaline, and possibly also DA, to modulate hematopoietic cell survival, proliferation, migration and engraftment ability [101]. Sympathoadrenergic control of thymic physiology and activity has also been described [39,102].

Upon stimulation, SNS terminals in immune organs can store, take-up and release norepinephrine [103], which interacts with the adrenoceptors expressed on target immune cells. The splenic norepinephrine content may be 95% depleted after application of the toxin 6-OH-dopamine (6-OHDA), indicating that most of the norepinephrine is of neural origin [99]. DA release was also demonstrated in the spleen [104]. Considering there is no dopaminergic innervation of this organ, it seems that DA could be taken up from the plasma by sympathetic terminals, partially converted into norepinephrine and released in response to neural activity. Indeed, plasma DA levels increase after exposure to a variety of stressors capable of enhancing sympathetic tone [105]. As has been extensively reviewed [39], once they reach leukocytes, catecholamines affect the trafficking, circulation, proliferation and the production of cytokines by different immune cells [23,102]. Biphasic immunomodulation of catecholamines has been described depending on the context, the amount of transmitter released and the receptors activated [106]. Thus, by stimulating β_2_-adrenoceptors, norepinephrine and epinephrine can inhibit the production of proinflammatory cytokines, including TNF-α, IL-12 and INF-γ released by antigen-presenting cells and Th1 cells, while stimulating the production of immunosuppressive cytokines like IL-10 and TGF-β [102,107]. Through this mechanism, endogenous, systemic catecholamines can selectively suppress Th1 responses and cellular immunity, provoking a shift towards Th2 dominance of humoral immunity. Alternatively, catecholamines can stimulate the immune response under certain conditions by inducing proinflammatory cytokines like IL-1β and TNF-α, and chemoattractant factors like IL-8 that acts on α2-adrenoceptors expressed by monocytes and macrophages, promoting the synthesis of TNF-α and other cytokines [108,109]. Thus, SNS activation during an immune response could serve to localize the inflammatory response by inducing the recruitment and activation of neutrophils, and/or other immune cells. At the systemic level, this activation suppresses the Th1 response in order to protect the body from the harmful effects of proinflammatory cytokines and other products released by activated macrophages [102,110].

#### 2.1.2. How Psychostimulants Can Modulate Catecholamine Levels

Psychomotor stimulants produce behavioral changes accompanied by enhanced alertness, excitement and motor activity, which has mainly been explained by their effects on catecholaminergic neurotransmission. DAT, VMAT and MAO are targets of amphetamines, which by molecular analogy compete with DA for binding to these proteins, while the classic effect of cocaine is to block the DAT [111,112]. Interestingly, it was recently demonstrated that the inhibition of DA reuptake by cocaine is actually mediated by autophagic degradation of DAT, but not of serotonin transporters in the nucleus accumbens (NAc), modulating the behavioral effects of cocaine [113]. Consequently, cocaine and amphetamines produce a common increase of DA in the synaptic cleft, and they enhance catecholaminergic transmission, although their influences on reuptake mechanisms slightly differ. Thus, the behavioral consequences of psychostimulant exposure have generally been attributed to their ability to elevate DA levels in the mesocorticolimbic system and particularly, in the NAc [114,115,116]. Psychostimulants also exert sympathomimetic properties that seem to be due to the effects of these drugs on postganglionic noradrenergic terminals in the SNS. This noradrenergic activation is responsible for an increase in blood pressure at a peripheral level and also for the higher levels of vigilance due to its central effects on the locus coeruleus [117].

Based on their effects on DAT and catecholamine levels, psychostimulants could affect immunity through at least three different pathways: (1) increased central DA, which modulates communication from the brain to the immune system; (2) increased DA and norepinephrine in the peripheral SNS, which influences immune cells; and (3) increased autocrine/paracrine DA release from immune cells. With regards to the first pathway, a central stimulus like stress that also modulates dopaminergic transmission can influence the immune response by activating both the SNS pathway and the HPA axis [118]. Morphine can also modulate immune function through both of these pathways [44,119], although information regarding psychostimulants is scarce at this level [120]. As mentioned above, psychostimulant drugs mainly affect the mesocorticolimbic dopaminergic system. Within the limbic system, the NAc [121] and the amygdala [122] have been implicated in the modulation of certain peripheral immune responses. It is possible that this modulation could be due to limbic connections with the SNS and HPA axis. Data provided by 6-OHDA lesions in the NAc, which affect ventral tegmental area neurons through retrograde transport, demonstrated the participation of this central pathway in the immunomodulatory effects of amphetamine [65]. The loss of dopaminergic terminals on NAc neurons not only reverses the peripheral effects of amphetamine on the lymphoproliferative splenic response, but also, it blocks the increase in met-enkephalin levels in the spleen and prefrontal cortex. Since the lesions involved affect ventral tegmental area neurons, these dopaminergic neurons may also project to other brain areas, such as the prefrontal cortex and amygdala. In turn, these areas send GABAergic and glutamatergic projections to the ventral tegmental area [123]. The pathway transmitting the messages evoked by amphetamine from the mesolimbic system to the immune system was not assessed in this study, although a possible link between the NAc and the extended amygdala in emotional control has been proposed [65]. The modulation exerted by the amygdala (and hence by the NAc) on the SNS could reach the spleen and adrenal glands [121,122], and may therefore explain this phenomenon. This central modulation of the spleen through the SNS was also recently demonstrated via the activity of B-cells in a T-dependent immune response [97].

With regards to the other two pathways, since the catecholamine machinery targeted by psychostimulants is expressed by immune cells and at SNS terminals, these drugs may act directly on these peripherally expressed dopaminergic and noradrenergic systems. Thus, in addition to the messages triggered by central dopaminergic activation, these drugs could increase catecholamine in situ in the peripheral cellular milieu. Indeed, the immune cells and the SNS terminals are reached by the drugs before they cross the BBB, and these effects might precede and influence central processes. We have evidence that blocking D1- and D2-type DARs through intraperitoneal (i.p.) administration of selective antagonists reverses the effects of amphetamine on the immune response [65]. Hence, it is possible that this antagonism involves central, as well as peripheral, dopaminergic receptors acting on immune cells and sympathetic synapses.

Other indirect evidence has been obtained from our recent data regarding adaptive peripheral immunity and a possible involvement of peripheral D_5_ DARs in the behavioral response to cocaine [124]. We used the paradigm of drug-seeking behavior in a model based on Fischer 344 (F344) and Lewis rats, which have different immune cell profiles and distinct sensitivities to the reinforcing effects of cocaine, F344 rats being more resistant to relapse. We transferred bone marrow from Lewis to F344 rats (F344/LEW-BM rats) and we observed a shift in their immune cell profile, as well as in their behavioral response to cocaine, resembling that of Lewis donor rats rather than that of the control group transplanted with F344 bone marrow (F344/F344-BM rats). Thus, only those F344 rats that received Lewis bone marrow cells reinstated cocaine-seeking behavior, and these rats also had fewer peripheral CD4^+^ CD25^+^ T-cells and immunosuppressive cytokines (TGF-β), together with higher proinflammatory cytokines (IL-17A) and D_5_ DARs in their spleens. We propose that the stronger D1-type inhibitory tone of Lewis-derived cells could inhibit Tregs and suppress their release of immunosuppressive cytokines (inhibition of inhibition), facilitating a Th17 response to cocaine. We also propose that these Th17 CD4^+^ T-cells mediate long-term immune memory to the drug, whereby they are able to recognize cocaine and be activated by drug re-exposure. This immune memory might be suppressed in F344-derived cells due to their weaker D1-type tone [124]. Futures studies are needed to better understand the peripheral catecholaminergic modulation of immunity that might even precede the central effects of psychostimulants.

### 2.2. Enkephalins

#### 2.2.1. Enkephalins as Neuro- and Immune-Modulators

Enkephalins are all derived from the same precursor molecule, proenkephalin, a 245 amino acid protein that contains four copies of met-enkephalin within its sequence [125]. To generate biologically active peptides, proenkephalin must undergo post-translational processing, such as cleavage by the prohormone converting enzymes PC1 and PC2, or plasma and tissue kallikrein that also selectively cleave proenkephalin [126,127]. This enzymatic processing is tissue-specific and gives rise to different final products depending on the capacity of each tissue to express distinct enzymes [128]. Moreover, the final effect of the opioid peptides will also depend mainly on the local expression of opioid receptors [129], classified as MORs, DORs and κ receptors [130]. Met-enkephalin shows high affinity for DORs and MORs, and as the latter predominate in the NAc, the main met-enkephalin activity in this brain area is mediated by them [131]. The expression of opioid receptors by immune cells has recently been reviewed [132], and it is noteworthy that inducible expression of DORs and MORs was described in rat spleen lymphocytes following Con A stimulation [133]. Met-enkephalin also interacts with the opioid growth factor receptor to delay passage through the G_1/S_ interface of the cell cycle and, as such, it is considered an important homeostatic regulator that influences the onset and progression of autoimmune diseases and cancer [134]. The proenkephalin gene itself is expressed in the CNS and in non-neural tissues of mesodermal origin during organogenesis [135], with its expression diminishing as cell differentiation proceeds. A significant amount of proenkephalin protein has been detected in astroglia and lymphocytes [136], which was located in the nucleus and had its functionality described [137].

Regarding met-enkephalin, its release is dependent on a cell’s ability to express the cleavage enzymes, with PC1 and PC2 present in neurons but not astrocytes. However, astrocytes are known to release unprocessed proenkephalin and at the same time, they release carboxypeptidase E that can cleave it, suggesting the existence of extracellular enzymatic processing [138]. It is therefore possible that part of the biological activity of opioids could be exerted by the intact proenkephalin protein or other precursor-derived peptides containing the met-enkephalin sequence that are processed extracellularly. In the rat spleen, since cells of the myeloid lineage express PC1 and/or PC2 (monocytes and neutrophils), they are the only immune cells able to produce met-enkephalin [136,139,140,141,142]. Proenkephalin mRNA transcripts have been detected in T-cells and mononuclear cells from the bone marrow, thymus and spleen, as have cryptic met-enkephalin-containing peptides derived from this protein [136,143]. Furthermore, stimuli such as those induced by LPS (lipopolysaccharide) and Con A enhance met-enkephalin release [136,142]. As already mentioned, many cell types cannot process proenkephalin to active opioid peptides, including lymphocytes, although larger precursor peptides containing the met-enkephalin sequence could be cleaved extracellularly by PC enzymes released by macrophages. Thus, the versatility of the enkephalinergic system is sustained by: (1) the existence of several molecules, from the precursor protein to the numerous derived peptides; (2) the variety of cells from different systems that express proenkephalin mRNA, including neurons, astrocytes, lymphocytes, macrophages and neutrophils; (3) the fact that there are several ways to process the precursor protein in order to release different peptides with known and unknown opioid activity; and (4) the variety of receptors that can be activated on different target cells.

Although opioid modulation of the immune response is not fully understood, dampening proenkephalin gene expression increases the proliferation of splenocytes stimulated with Con A [144]. Moreover, proenkephalin also mediates the apoptosis activated by cellular stress through the transcriptional repression of genes like NF-κB and p-53 [145]. Likewise, met-enkephalin synthesized by nervous and/or endocrine tissue has been implicated in stress-induced immunosuppression [139,146,147]. Immunosuppression was also described in morphine users, as well as in experimental and in vitro models [148], although other mechanisms have been associated with morphine administration that are not shared by opioid peptides [149]. Nevertheless, there is evidence of a proinflammatory role for this peptide, since it is able to induce bone marrow-derived dendritic cells to polarize predominantly to a myeloid subtype [150], and to also induce M2 macrophage polarization to an M1 phenotype [151], favoring a Th1 response. The upregulation of MHC class II expression and that of key co-stimulatory molecules on these antigen-presenting cells was also observed following met-enkephalin treatment [150,151]. Met-enkephalin was seen to have antitumor effects by balancing the immune response suppressing myeloid-derived suppressor cells and enhancing T-cells through a mechanism blocked by naltrexone [152]. These antitumor effects might correlate with the suppression of inflammation, further evidence supporting the use of met-enkephalin in adjuvant therapy for tumors. Met-enkephalin was shown to inhibit influenza infection, and it has been proposed as a possible therapeutic agent for cancer and as an adjuvant in vaccine preparation [150,153,154,155]. Additionally, an increase of proenkephalin was recently reported in Tregs that maintain skin homeostasis and act in wound healing [156]. Thus, it is possible that met-enkephalin acts as a biphasic modulator of the immune response depending on the concentrations reached and/or the main receptor activated [157]. This bell-shaped dose-response curve was also described a few decades ago for other endogenous opioids peptides [158] and it indicates the involvement of multiple receptors that transduce bidirectional signals that produce opposing biological effects. Thus, met-enkephalin at low concentrations seems to be proinflammatory, whereas at high concentrations it induces immunosuppression, a result of either MOR or DOR activity, respectively [132,157,159].

#### 2.2.2. How Psychostimulants Modulate Enkephalin Levels

In addition to the release of mesocorticolimbic DA, primarily involved in the initial action of psychostimulants, opioid neurotransmission in the NAc shell influences the hedonic impact of these drugs and that of natural reinforcers [160,161]. Intra-NAc administration of a DOR antagonist was seen to diminish the self-administration of cocaine, while intra-ventral tegmental area administration enhances this behavior [162]. We demonstrated that cocaine can provoke an increase in proenkephalin gene expression in key mesocorticolimbic areas following a time-dependent course, irrespective of whether the drug was self-administered or not [163]. In addition, previous studies in our lab demonstrated differential proenkephalin gene expression in F344 and Lewis rats [164,165]. As mentioned above, these rats show distinct vulnerabilities to the reinforcing effects of several drugs of abuse, including cocaine and morphine. Thus, F344 rats express higher basal levels of proenkephalin mRNA in the NAc and dorsal striatum, and they have greater functionality of MOR than Lewis rats, the latter being more vulnerable to the reinforcing effects of cocaine [164,165]. Increased opioid receptor expression and that of the protein precursors of opioid peptides was also evident in certain brain areas following cocaine self-administration [166]. Indeed, studies of proenkephalin KO mice highlighted that the enkephalinergic system is involved in regulating the long-lasting neuroadaptations in the NAc that underlie behavioral sensitization to cocaine [167].

Regarding the immune system, the synthesis and release of proenkephalin and its derived peptides (including met-enkephalin) was described following exposure to amphetamine in immune organs (spleen, thymus and bone marrow) and simultaneously in mesocorticolimbic areas (NAc and prefrontal cortex) [66]. In addition, we observed the release of cryptic met-enkephalin peptides in the supernatant of nonactivated cultured splenic mononuclear cells from amphetamine-treated rats, and at similar levels to those induced by Con A stimulation. Interestingly, the amphetamine treatment involved i.p. administration of only one dose, five days before splenocyte culture [66]. These data suggest that similar pathways may be triggered by amphetamine and Con A in immune cells, which could be explained by recent evidence that methamphetamine is a TLR4 agonist (see below) [168]. As far as we know, the simultaneous enkephalinergic activation in immunocytes and in the brain following exposure to amphetamine is the only evidence of a drug of abuse inducing similar biological changes in both systems. Since no further studies were carried out regarding other drug effects occurring simultaneously at both a central and immune level, we strongly recommend that such studies be undertaken to transcend the belief that these drugs only affect the brain.

### 2.3. Toll-Like Receptors (TLRs)

#### 2.3.1. TLRs as Immune and CNS Receptors

TLRs are pattern-recognition receptors that recognize pathogen-associated molecular patterns from microorganisms or danger-associated molecular patterns from damaged tissue. The first TLR to be identified was IL‑1R type 1 (IL‑1R1), cloned in 1988, the cytosolic domain of which has homology to the cytosolic domain of the *Drosophila melanogaster* protein Toll. In the next decades, more than a dozen Toll homologues were found (i.e.,: TLRs) and characterized as pattern-recognition receptors for bacterial, viral and parasitic products [169]. Of these, TRL4 has been implicated in LPS recognition and TLR2 in sensing bacterial lipopeptides, heterodimerized with TLR1 to recognize triacylated lipopeptides, or with TLR6 to recognize diacylated lipopeptides and other non-lipopeptidic pathogen-associated molecular patterns [170]. However, endogenous ligands have also been described for these receptors, and TLR4 and TLR2 can sense endogenous agonists like heat-shock proteins and high-mobility group box-1 protein [171,172]. More recently, saturated fatty acids have also been described to be TLR4 ligands [173]. Once activated, both TLR4 and TLR2 are able to stimulate the myeloid differentiation primary response 88 (MyD88)-dependent signaling pathway, also used by other TLRs [174].

The expression of TLRs by innate immune cells was originally described, providing a link between them and adaptive immune cells. Beyond their ability as pattern-recognition receptors involved in defense mechanisms, these receptors are currently the subject of intense research since they have been described in other cells, and they recognize several other endogenous and exogenous ligands. This versatility extends their activities to other physiological or pathological processes, such as pain transmission or cerebral ischemia [175,176,177]. TLR expression among resident CNS cells was first studied in microglia (an innate immune cell) and astrocytes, with neurons only considered targets of glial factors released after TLR stimulation [178,179]. However, TLR4 and TLR2 expression by neurons was later described, as was their modulation by IFN-γ, similar to that in immune cells [176,180,181]. Hippocampal TLR4 signaling participates in neuroinflammatory responses associated with traumatic brain injury [182], seizures [183], imbalances in excitatory/inhibitory strength [180] and long-term potentiation, defects that affect hippocampal memory [184,185]. TLRs have also been shown to participate in neurogenesis and neuronal differentiation, and a growing body of evidence implicates these receptors in other CNS mechanisms [186].

#### 2.3.2. How Psychostimulants Can Modulate TLR Activation

Some of the first evidence that TLRs are involved in the effects of psychostimulants emerged from HIV research when it was demonstrated that methamphetamine down-regulates TLR9 expression [187]. Other drugs of abuse had been proposed as TLR2 and TLR4 agonists, such as morphine and alcohol, and as modulators of their expression by macrophages and/or glial cells [188,189,190,191,192]. This evidence laid the groundwork for new theories on addiction, proposing the activation of innate immune genes in the CNS as a key mechanism for the addictive process [76]. These findings were later extended to the effects of psychostimulants. Thus, in addition to its action on the DAT, cocaine was shown to act as a TLR4 agonist in the CNS, and this signaling is necessary for the cocaine central rewarding effects [193].

A molecular interaction between cocaine and the TLR4/MD-2 complex was demonstrated, showing that cocaine docked to the same binding domain of MD-2 as LPS, the classical TLR4 agonist. Likewise, cocaine increased IL-1β expression by isolated neonatal microglial cells in vitro and by ventral tegmental area neurons in vivo, possibly in a TLR4-dependent manner. This TLR4-mediated enhancement of IL-1β expression contributes to the increase in DA in the NAc and it participates in the reinforcing effects of cocaine [193]. Cocaine was later proposed as a modulator of TLR2 expression by cultured microglial cells, concomitant with microglial activation [194]. Indeed, the reactive oxygen species (ROS)/endoplasmic reticulum (ER) stress-ATF4 pathway was shown to underlie the cocaine-mediated microglial activation, involving TLR2 upregulation. In addition, it has been previously demonstrated that cocaine exposure induced autophagy in microglial cells, which involved upstream activation of two ER stress pathways (EIF2AK3- and ERN1-dependent) [195]. Interestingly, a very recent study showed that cocaine induces autophagic degradation of DATs in the NAc, a process that influences the cocaine-conditioned place preference behavior of mice [113]. Considering cocaine is an agonist of TLR4, which can activate ROS-mediated autophagy in several cell types [196], we suggest that these new roles described for cocaine could be explained by TLR4-induced autophagy in CNS cells.

It also appears that cocaine-induced microglial activation is driven by TLR signaling. Cocaine downregulates miR-124 and it activates microglia by targeting Krüppel-like factor 4 (KLF4), a transcription factor lying downstream of the TLR4 pathway, as well as acting through other molecules involved in TLR4 signaling, including MyD88, IRAK1 and TRAF6 [197]. In addition, miR-124 overexpression significantly blocks the cocaine-mediated upregulation of M1 proinflammatory markers (TNF-α, CCL2 and NOS2), and it enhances the expression of M2 anti-inflammatory mediators (TGF-β, IL-4 and IL-10) in microglial cells exposed to cocaine [197].

Alternatively, TLR4 deficiency has been linked to deficits in low-frequency stimulation-induced NMDAR-dependent long-term depression in the NAc core, concomitant with an attenuation in drug reward learning in the conditioned place preference paradigm [198]. Intra-ventral tegmental area instillation of a TLR4 antagonist produced a reduction in the reinstatement rate when animals trained to self-administer cocaine were re-exposed to this drug [199]. Moreover, increased levels of TLR4 and other immune-related proteins were detected in the striatum of mice that self-administered cocaine [200]. All this evidence reinforces the possible implication of TLR4 signaling in the rewarding properties of cocaine, suggesting that this may serve as a potential therapeutic target for the treatment of addiction. However, although this TLR4 activation in resident brain cells following psychostimulant exposure could also occur at the peripheral immune level, this has yet to be explored. As far as we know, the only data regarding the influence of cocaine on peripheral TLR expression come from our latest work [124]. Paradoxically, we found weaker splenic TLR4 and TLR2 mRNA expression when F344 rats were transplanted with Lewis bone marrow, as well as lower levels of IL-1β, indicating more limited innate immune activation in the rats that reinstated the cocaine-seeking behavior. In parallel, we found higher levels of IL-17A in spleen cells derived from Lewis bone marrow, leading us to propose an “adaptive immune signaling” hypothesis that could underlie some key stages of the addictive process, and that involve CD4^+^ T-cells undergoing Th17 polarization.

Although our hypothesis is not mutually exclusive to the innate immune theory, as both innate and adaptive immunity complement each other, and we only studied peripheral markers, our data do suggest that a balance towards an innate immune response would predominate over the adaptive response in F344 rats that are resistant to relapse. By contrast, the vulnerability to relapse would be associated with adaptive immunity predominating over innate signaling, and immune mediators released by T-cells may trigger this behavioral response, making rats transplanted with Lewis cells prone to relapse.

It should be noted that both adaptive and innate mechanisms at peripheral and central levels, as well as the neurobiology that has already been described to underlie addiction, are parallel processes that act synergistically, and that might be relevant in different stages of the whole addiction process. In particular, we demonstrated a key role of peripheral adaptive signaling in cocaine relapse, a process in which long-term memory of the drug is essential and that can be sustained by immunological memory. While comprehensive understanding of these processes remains to be achieved, pharmacological manipulation of the peripheral immune response opens up novel therapeutic opportunities to treat these disorders.

## 3. How Psychostimulants Can Influence HIV Infection

The advent of ART has enabled people living with HIV to confront this situation as a chronic illness more than a fatal disease. However, HIV is not completely eradicated by ART and it persists in specific cells or tissues that act as reservoirs where viral replication continues under different conditions [201,202]. HIV can access the CNS, in part through the transmigration of infected peripheral monocytes that differentiate into macrophages and that can infect other brain resident cells. These long-living infected cells act as a central viral reservoir despite ART, and they contribute to inflammation and neuronal damage.

HAND is reported in a substantial percentage of people living with HIV and treated with ART. Considering the high prevalence of drug abuse among these individuals, efforts have been made to understand the possible connection between drug use and HAND [74]. Significant effects in the peripheral immune system have been reported in psychostimulant users [203,204] and in experimental models [64,65,205]. Thus, beyond HAND, understanding these peripheral mechanisms could shed light on the influence that drugs of abuse can exert on the immune surveillance of people living with HIV, which may ultimately affect the evolution of infection even in the presence of ART.

To better characterize the immune mechanisms affected by psychostimulants in association with HIV progression, it would be helpful to consider the tools the immune system uses effectively against HIV. This information can be partially obtained from the exceptional cases of “elite controllers” and “post-treatment controllers”, individuals who control infection (analogous to functional cures) without or after ART treatment, respectively [202,206,207]. Distinct and varied mechanisms are involved in both cases, with a sustained and highly potent HIV-specific CD8^+^ T-cell and natural killer (NK) response in elite controllers [208,209], and a weaker activation of T-cells, particularly CD4^+^ cells, in post-treatment controllers [207]. Although most studies assume that the mechanisms that achieve control are the same as those that maintain the virus under control, this assumption has been challenged and it was proposed that the optimal immune response differs in both scenarios [210]. Elite controllers are associated with the protective HLA alleles (HLA-B57 and HLA-B27), implying an immunodominant host CD8^+^ T-cell response during the earliest stages of the infection (acute phase). However, a less inflammatory response was observed among post-treatment controllers, which is consistent with the idea that the activity needed to initially control infection would probably be stronger than that needed to control an ART-induced lower viremia. It is also important to note that early initiation of ART prevents the establishment of a large virus reservoir, yet this cannot be so early as to prevent the generation of memory T-cells. Thus, a limited temporal window of opportunity was described as a key factor for ART success. Other remarkable aspects to the control of HIV have been derived from the different immune responses in women, men and pediatric patients, as well as in the exceptional elite controllers and post-treatment controllers that experience a rebound (for a detailed review see Goulder and Deeks, 2018).

Considering the low percentage of elite controllers and that most people living with HIV are subjected to chronic ART, most data have been obtained from post-treatment controllers, suggesting the aspects of the immune response that would be desirable to activate in these people living with HIV. Thus, a potent anti-HIV response should theoretically be more appropriate to achieve initial control of infection, whereas a less proinflammatory response may be better to maintain control of infection under ART. For the purpose of this review, we will analyze whether psychostimulants might achieve precisely the opposite of what is desirable at the peripheral immune level, making it more difficult to resolve the infection, and highlighting the participation of catecholamines, enkephalins and TLRs. We have already covered how these mediators influence immunity and how psychostimulants influence them. As such, the interference that cocaine and amphetamines may exert on these neuro-immune messengers at a peripheral level may ultimately affect HIV, as could be partly deduced from the data already presented. Thus, we will describe some studies that have examined the possible interplay between these elements in immunocytes.

### 3.1. How HIV Infection Can Be Modulated by Catecholamines Released by Psychostimulants

Prior to ART, DA-rich brain areas (such as the striatum, prefrontal cortex and substantia nigra) seemed to be particularly vulnerable to HIV infection, putting the focus on this transmitter as a factor associated with development of HAND. Many studies and revisions have focused extensively on the influence of CNS DA in HIV neuropathology, even in association with drugs of abuse [74,211]. However, the possible influence of psychostimulants on the catecholaminergic tone on peripheral immune cells has not yet been addressed, except in some in vitro studies. Thus, without reiterating the current evidence excellently reviewed elsewhere, we first propose that similar mechanisms to those described for the central dopaminergic system in a drug user’s HAND should be extended to the effects of the dopaminergic system on peripheral immune cells. Secondly, we will address the in vitro findings indicative of immune dopaminergic modulation under the influence of psychostimulant use that may affect peripheral immunity, and we can speculate on their physiological significance.

Initial evidence indicated DA and its D1-type receptors are directly implicated in the effects of psychostimulants on immune cells that may influence HIV infection [212]. Researchers found that DA upregulates CCR5 HIV co-receptor expression through monocyte-derived macrophages after methamphetamine treatment, consequently enhancing the rate of HIV infection. Additionally, they showed D1 receptor mRNA in macrophages to be identical to that expressed by neurons and consistent with their functionality [212]. Similar data were later reported regarding the involvement of both types of DARs in the upregulation of CCR5 and CXCR4 expression in monocyte-derived dendritic cells by methamphetamine, along with enhanced HIV infection [213]. Methamphetamine had similar effects on CCR5 expression by uninfected microglia, yet this was higher after simian immunodeficiency virus (SIV) infection [214]. CCR5 upregulation was also reported in human mononuclear cells after exposure to cocaine [215], as well as in peripheral blood leukocytes [216] and CD4^+^ T-cells [217], accompanied by a higher viral load in all cases. Although no dopaminergic modulation has yet been explored, in the light of the existing data, it is very likely that DA also mediates these effects of cocaine. Indeed, a role for DA in CCR5 upregulation has been confirmed [218] and D_4_ receptors were seen to mediate cocaine-induced enhancement of HIV infection in quiescent CD4^+^ T-cells [219], probably mediated by CCR5 upregulation. Thus, exposure to psychostimulants allows the virus to more efficiently infect its peripheral target cell populations and replicate. It is important to point out that most of these data come from in vitro studies that depend on the autocrine release of DA from cultured cells, with no DA coming from other sources (such as other immune cells or SNS nerve terminals) that may be released as a paracrine or systemic response to psychostimulant exposure. Considering these in vitro limitations, it is possible to suggest an even stronger effect after in vivo exposure to psychostimulants.

Beyond the increased possibility of HIV infection, it is interesting to analyze the functional relevance of a “chemoattractant-like” function of DA, considering its ability to upregulate chemokine receptors that will, in turn, modulate leukocyte trafficking. Following three weeks of intravenous (i.v.) cocaine self-administration, we observed striking splenomegaly [124], as had been described previously [64,220]. This cocaine-induced effect on the spleen can be explained in several ways (see Assis et al., 2020), although it might involve the chemoattractant-like effect observed for DA, since the spleen is one of the most densely innervated SNS organs [221] and cocaine may target the DAT expressed on splenic cells. Further research will be needed to tease out the biological significance of the DA induced upregulation of chemokine receptors over and above the unwanted effect on HIV-enhanced entry into target cells.

In addition to the direct effect on chemokine receptors, dopaminergic tone could indirectly affect other immune mechanisms that also influence HIV infection, such as the “inhibition of inhibition” on immunity triggered by D1-type DARs on Tregs [36,95]. We recently demonstrated a distinct genetic vulnerability in cocaine-induced relapse, probably mediated in part by altered splenic D_5_ dopaminergic receptor expression, concomitant with different levels of Treg cells [124]. As a result, we proposed a possible target of cocaine in terms of the dopaminergic modulation that inhibits immunosuppressive Treg activity (lower IL-10 and TGF-β levels), differentially modulated according to genetic variance in the expression of D_5_ dopaminergic receptors. A less immunosuppressive environment, accompanied by the higher IL-17A levels following cocaine re-exposure, could also be relevant to HIV infection, as it is desirable to trigger a less proinflammatory response in people living with HIV, which can be hampered by cocaine use. Interestingly, enhanced IL-10 levels have been described in Con A-stimulated splenocytes following cocaine self-administration, together with higher TNF-α levels [64]. However, after an extinction period followed by drug re-exposure, reduced IL-10 levels were also found, in consonance with our findings. Thus, during a period of daily i.v. cocaine administration, opposing forces seems to be induced as the release of both anti- and proinflammatory cytokines was favored (IL-10 and TNF-α, respectively). Nevertheless, after cocaine re-exposure, a weaker anti-inflammatory response dominates: less IL-10 and more INF-γ in the earlier study; and from our data, less IL-10 and more IL-17A. Elevated IL-10 was also observed in cultured macrophages exposed to cocaine, yet along with increased HLA-DR expression that reflects macrophage activation [222]. Again, opposing forces coexist under the influence of cocaine. These results highlight a critical issue related to the patterns of drug consumption and the moment in which the immune variables are analyzed, a topic that will be discussed below. Further experiments will be necessary to more fully understand the relevance of the neuro-immune adaptations on the peripheral and central dopaminergic system, which might affect HIV progression in conjunction with psychostimulant use.

### 3.2. How HIV Infection Can Be Modulated by Enkephalins That Are Released by Psychostimulants

Over the past decades, considerable evidence has accumulated of an influence of morphine on immunity, with combinatorial effects observed between its use and HIV infection [223]. Most data indicate an immunosuppressive effect of morphine, while others suggest inflammatory effects that mainly influence the progression of HAND. Most of the immunosuppressive effects of morphine have been attributed to opioid receptor activation [148]. We have already indicated that psychostimulants can profoundly affect the expression of proenkephalin and the release of met-enkephalin, mostly from studies on the brain but also from those focusing on immune cells [66,163,167]. Thus, considering that morphine like met-enkephalin acts on MORs and DORs, it may be inferred from the findings with morphine that this endogenous opioid peptide could influence the immunocompetent state of people living with HIV. Since morphine has also been described as an agonist of TLR4 [149], it is necessary to clearly differentiate which of its effects are strictly mediated by opioid receptors. Thus, we will evaluate the studies in which the effects of morphine on HIV progression can be reversed by antagonists of opioid receptors or they are abrogated in MOR KO animals, as well as the effect observed in the presence of selective MOR and/or DOR agonists.

Interestingly, early studies regarding morphine and HIV or SIV demonstrated interactions of opioid receptors with chemokine receptors, such as those of MORs with CCR5, due to direct protein–protein dimerization/oligomerization at the surface of immune cells [224] and astroglia [225]. In addition, there may be cross-desensitization between the receptors when they form a heterodimer [226,227]. CXCR4 and DOR also form heterodimeric complexes, with possible inactive conformations unable to become active when they are simultaneously stimulated by their ligands [228]. Furthermore, the activation of MORs upregulates CCR5 and CXCR4 expression in CD3^+^ lymphoblasts and CD14^+^ monocytes [229], as well as that of CXCR4 in bone marrow progenitor cells [230]. MORs have also been implicated in the enhancement of HIV Tat-induced neurodegeneration of glial cells [231]. However, when CCR5 was lost from glia, paradoxically, morphine appeared to protect neurons from Tat-induced toxicity [232], suggesting glial CCR5 mediates the neurotoxicity of Tat that is potentiated by MOR activation. In addition, morphine-mediated defective autophagy has been linked to MOR signaling in astrocytes, interacting with the ER stress/autophagy axis [233]. As we proposed previously, all the phenomena from opioid receptors expressed on resident CNS cells could be extended to peripheral immune cells.

Direct evidence of MOR activation on immune cells indicate that they promote naïve T-cell differentiation into Th2 cells [234], they downregulate IL-12p40 expression on macrophages during morphine withdrawal [235] and they inhibit phagocytosis [236]. This effect on phagocytosis was also evident after DOR and κ opioid receptor activation [236]. Thus, reduced cytotoxic activity of Th1 cells and phagocytes, in addition to increased expression of HIV co-receptors (CCR5 and CXCR4), may mediate the deleterious effects observed following opioid receptor activation. Considering that met-enkephalin is a MOR and DOR agonist, the increased release of this peptide that is induced by psychostimulants might mediate all these aforementioned actions, compromising HIV infection.

One interesting issue is that the heterodimeric opioid and chemokine receptors complexes, such as the MOR-CCR5 and DOR-CXCR4 dimers, may be related to the lack of effectiveness of some ART drugs. Co-exposure with morphine negates the effects of maraviroc, a drug that inhibits HIV entry acting on CCR5 [237]. Thus, a similar effect on the efficacy of maraviroc might be generated by the psychostimulants due to the met-enkephalin they release and also considering that DA upregulates CCR5.

In terms of the effects of psychostimulants on HIV infection, there would appear to be an intricate interplay between DA, met-enkephalin and TLR4 (see above). The upregulation of TLR4 by met-enkephalin in bone marrow-derived dendritic cells appears to be mediated by the activation of MORs but not DORs [238]. Interestingly, combined exposure of Tat and morphine enhances MOR expression on microglial cells [239], and that of TLR4 [240]. In light of the evidence [238], this upregulation of TLR4 might also be mediated by MOR activation. Alternatively, we described a dopaminergic influence in the increase in met-enkephalin induced by amphetamine in certain brain areas and in immune cells and tissues [65]. Thus, related effects of psychostimulants on DA, met-enkephalin and TLR4 at the CNS and peripheral immune level might at least partially underlie some of the effects observed in people living with HIV that use cocaine and/or amphetamines. These effects appear to involve CCR5 and CXCR4 upregulation, as well as a possible synergistic action of Tat on them.

### 3.3. How HIV Infection Can Be Modulated by TLRs Activated by Psychostimulants

Although CD4^+^ memory T-cells are considered the major HIV reservoir, macrophages also contribute to HIV persistence in addition to their role in HIV production. Indeed, tissue resident macrophages in the lymph nodes, lung, liver, adipose tissue, gastrointestinal and genitourinary mucosae, and microglia in the CNS serve as tissue reservoirs for HIV [241]. As mentioned previously, TLRs are mainly expressed on innate immune cells but also, to a lesser extent on T-cells, such that the TLR4 modulation described by psychostimulants [168,193] could affect both HIV reservoirs: macrophages and CD4^+^ T-cells.

Among TLRs, those that can recognize viral structures and activate an immune response are mainly present in endosomal membranes: TLR3 recognizes double-stranded RNA [242], while TLR7 and TLR8 recognize single-stranded RNA [243] and TLR9 recognizes unmethylated double-stranded DNA [244]. Moreover, activation of TLR3, TLR7 and TLR9 could help HIV vaccine strategies and/or ART, since these receptors reactivate HIV production and improve the functioning of immune cells that target latently infected cells, reducing the HIV reservoirs [245,246,247,248]. As well as these endosomal TLRs, TLR10 at plasma membrane recognizes gp41 and other HIV proteins, in conjunction with TLR1 and TLR2. HIV infection was associated with stronger TLR10 expression, indicating that this TLR might play an important role in HIV infection and pathogenesis [249]. Furthermore, p17, p24 and gp41 HIV structural proteins bind to TLR2 [250].

Regarding the action of psychostimulants on TLRs, methamphetamine is known to downregulate TLR9 expression on macrophages and suppress host innate immunity against HIV infection [187]. TLR4 activation and TLR2 modulation were further demonstrated as a consequence of psychostimulant use, although only in the CNS [168,193,194]. As far as we know, no studies have focused on the possible influence of psychostimulant-induced TLR4 activation in peripheral immune cells, particularly the macrophages and CD4^+^ T-cells that act as HIV reservoirs. Thus, we can only consider indirect evidence and speculate about the consequences of psychostimulant use in the evolution of HIV infection.

Studies regarding TLR4 activation by bacterial infection in people living with HIV may shed light on how a TLR4 agonist can influence HIV evolution. Early in vitro studies showed that a purified protein derived from *Mycobacterium tuberculosis* increased viral mRNA expression in HIV infected monocytes, although LPS failed to induce the detection of HIV RNA in these primary monocytes from 24 h to 5 days [251]. Bacterial LPS has been shown to activate TLR4 and induce HIV long-terminal repeat (LTR) transactivation [252], although other TLRs may also mediate this effect [253,254]. However, later studies demonstrated that repeated LPS exposure inhibits HIV replication, as witnessed by weaker expression and protein production of HIV p24, concluding that TLR4 ligands can induce tolerance and suppress HIV-LTR transactivation in human monocytic cell lines [255]. Along similar lines, the participation of TLR4 in HIV infection of macrophages in the genitourinary mucosa was recently reported, playing a key role in HIV sexual transmission and pathogenesis [256]. Since *Neisseria gonorrhoeae* augments mucosal transmission of HIV, both by inducing inflammation and by directly activating virus infection and replication, the interaction between commensal (*Escherichia coli*) or pathogenic (*N. gonorrhoeae*) bacteria on HIV latency has been studied in macrophages. Both these bacteria encode TLR2 and TLR4 ligands, and they repress HIV replication in macrophages, inducing a latency-like viral state. Soon after TLR4 stimulation, cellular mechanisms involving IRF1, NF-κB and HIV Tat lead to high levels of virus replication, whereas at later time points IRF1/IRF8 heterodimers repress HIV replication. Thus, productive HIV infection of macrophages seems to be altered by TLR signaling, with HIV activated by TLR2 and TLR5 signaling, and HIV replication repressed by TLR3 and TLR4. These data suggest that TLR4-mediated signaling (in this case induced by a microbe) represses HIV replication and induces viral latency in human macrophages, contributing to the establishment and maintenance of latent HIV infection in these cells. As such, these macrophages contribute to the viral reservoir in people living with HIV under ART [256]. These interesting data led us to suggest that chronic administration of other TLR4 agonists (e.g., cocaine or amphetamine) might also induce a similar effect on the TLR4 expressed by macrophages, microglia or even CD4^+^ T-cells. Such a phenomenon would have certain clinical implications for the establishment and maintenance of the latent HIV reservoir, both in the periphery and the CNS. Furthermore, the participation of TLR2 should also be considered in this process as cocaine enhances its expression by microglia [194].

### 3.4. Other Immune Factors That Can Mediate Psychostimulant-Induced Effects on HIV Infection

Without presenting a detailed description that would exceed the scope of this review, we will briefly mention some data indicating that cytokines and immune cells modulated by psychostimulants may influence the evolution of HIV infection, which may be linked to the events already described. There is evidence of innate immune proinflammatory markers in people living with HIV that use psychostimulants, as well as from in vitro models of HIV and drug co-exposure. Activated monocytes have been described in HIV-infected subjects that use methamphetamine, revealed by higher soluble CD14 levels, which are associated with the intensity of substance use [257]. Since soluble CD14 is involved in TLR4 activation under certain circumstances [258,259], an effect of methamphetamine on soluble CD14 levels due to TLR4 activation could be inferred. In addition, differential expression of genes associated with HIV latency, cell cycle regulation, innate and adaptive immune activation (mainly related to the TNF-α pathway), neuroendocrine regulation and neurotransmitter synthesis (including catecholamines) has been observed in blood from men infected with HIV that recently used methamphetamine [260]. Regarding cocaine, older adults with a cocaine use disorder had a higher neutrophil to lymphocyte ratio, described as a marker of chronic inflammation [261]. Methamphetamine increases TNF-α in human macrophages in vitro [262] and IL-6 in a human astrocyte cell line [263], and when combined with HIV gfp120 it increased TNF-α and IL1β production by rat microglia [264], also further increasing IL-6 in astrocytes [263].

Regarding adaptive immunity, there are some data related to IL-10, IL-17A and other cytokines, as well as CD4^+^ T-cell subsets. It was shown that regular methamphetamine use over 12 months in people living with HIV was associated with enhanced CD4^+^ T-cell activation and exhaustion [265]. Increased IL-6, INF-γ and CCR5 was observed following five days of i.p. cocaine administration in humanized mice, at which time co-exposure to HIV was initiated for two weeks. These mice had higher viral loads than HIV-infected animals that did not receive cocaine [217], suggesting that the initial cocaine-induced pro-inflammatory environment could favor HIV replication. In addition, the mice that were administered cocaine and HIV had more activated CD4^+^ and CD8^+^ T-cells (CD4^+^CD38^+^ and CD8^+^CD38^+^ T-cells), yet a weaker cytotoxic effector response in CD8^+^ T-cells. A key role for these cytotoxic cells in HIV control has been demonstrated by means of transient CD8^+^ T-cell depletion in SIV-infected macaques, which was associated with higher viremia [266,267] and was even more prominent in controller than in progressor animals [268]. Interestingly, we observed a higher CD4^+^/CD8^+^ T-cell ratio linked to vulnerability to cocaine relapse [124]. Thus, a clear proinflammatory signature accompanies psychostimulant exposure in people living with HIV, which ultimately may induce exhaustion of the adaptive immune response and/or weaken the cytotoxic response.

## 4. Concluding Remarks

The main objective of this review was to invite the reader to adopt a new perspective on the classical disciplines of neuroscience or immunology, and to try to contemplate the phenomenon of drug use and drug addiction, as well as HIV comorbidity, as integrated and holistic processes that do not respect the boundaries between these systems. Our intention was to contribute to our understanding of this complex game by analyzing classic players but in positions in which they are not often considered. Thus, we would like to break the praxis of contemplating the brain as the sole target organ for the action of psychostimulants or other drugs of abuse, assuming that the biological impact and the behavioral consequences of drugs are the result of mechanisms that operate across the whole body. The repercussions of drug use on immunity should no longer be considered as “collateral effects” but rather, as mechanisms activated and coordinated in conjunction with the CNS, the SNS, the endocrine system and perhaps other mediators of which we remain unaware of, as we recently demonstrated regarding the immune mechanisms underlying cocaine relapse [124].

In vitro studies isolate cell systems and challenge only a part of the whole, ignoring other mechanisms that may repress in vivo the findings found in vitro. Thus, it is highly relevant to always keep these limitations in mind and to rely on in vivo studies, either from animal models or clinical trials, which provide the best approximation to the real scenario. Thus, one factor that adds extreme complexity to our understanding of the biological bases underlying psychostimulant use and addiction are the patterns of use, which may lead to the emergence of adaptive mechanisms during periods of drug absence and subsequent re-exposition. Psychostimulant users frequently fluctuate between both phases of use and withdrawal, permanently challenging the body to reach states of allostasis that have been mainly studied in the CNS [269,270,271] but that may also operate in the immune system. These permanent attempts to achieve an equilibrium unleash a swirl of changes in an immune system that, in the case of people living with HIV, is also under challenge by a pathogen, HIV.

Among the adaptive mechanisms described in the CNS following psychostimulant exposure, sensitization (or inverse tolerance) and tolerance involve several molecular and cellular processes that can sometimes mask one another. Sensitization appears early in use, underlying the increasing interest in the drug and facilitating the acquisition of drug self-administration behavior, and constituting a relevant factor in the first encounters with the drug [272]. Thus, the importance of understanding this phenomenon lies in the potential role that sensitization has on the development of addiction, in recidivism after long periods of abstinence, and in the appearance of psychotic states induced by psychostimulants [271,273]. While tolerance appears later with sustained use, it coexists with the process of sensitization, but overlying it. However, during abstinence, tolerance mechanisms tend to disappear, while sensitization persists and emerges as a long-lasting mechanism that may induce relapse [271,274]. Accordingly, opposing forces may coexist in the CNS and predominate one over the other depending on the timing in the cycle of consumption. At the immune level, some anti- and pro-inflammatory conditions have already been described following cocaine exposure, withdrawal and drug re-exposure [64,124], and these may operate like those described for central sensitization and tolerance. Furthermore, several neuro-immune transmitters and/or modulators, such as those described here (e.g., catecholamines and enkephalins) undergo biphasic modulation, and they may induce opposing effects on immunity depending on the circumstances.

The immunocompetent state of an individual that uses psychostimulants may display a wide range of possibilities in terms of the pattern of consumption (from sporadic and intermittent use, to bingeing or chronic and sustained consumption), and with the possible simultaneous use of other drugs that could also affect immunity. Initial contact with HIV can arise at any time during this cycle, with immune scenarios that may influence how this incipient hit is coped with. Since HIV usually comes to stay and it persists beyond the initial response, the virus will be confronted with a dysregulated immune system that is subjected to the fluctuations imposed by subsequent drug re-exposure. In addition, psychostimulants could help HIV more easily infect host cells by directly upregulating its co-receptors CCR5 and CXCR4.

Without being able to ignore the current world situation, it is important to highlight that the breeding ground generated in an organism exposed to psychostimulants and HIV could constitute a “double jeopardy” for the coronavirus pandemic, as described recently [275]. This dual threat for people living with HIV that use drugs has multiple facets, ranging from immune dysfunction that might facilitate SARS-CoV-2 infection, to the chronic and uncontrollable stress implicitly associated with the pandemic that exacerbates psychiatric disorders and the use of drugs of abuse, as well as the danger to those who may be detoxifying to enter relapse. It is therefore of enormous relevance to unravel the effects psychostimulants have on peripheral immunity, which may involve neuro-immune mechanisms triggered simultaneously in the CNS and immune system, and that could provide us with a more global understanding of the harmful marriage between drugs and HIV.

## Figures and Tables

**Table 1 viruses-13-00722-t001:** The most relevant molecules classically associated with the immune system expressed by neural cells and classically associated with the central nervous system (CNS) expressed by immune cells.

Molecules Classically Associated with the Immune System that Are Expressed by Neural Cells	Molecules Classically Associated with the CNS that Are Expressed by Immune Cells
CCL-2 ^1^	Adrenocorticotropic hormone (ACTH) ^2^
CD3ζ	Arginine-vasopressin ^2^
Complement system (C1q, C3)	Atrial natriuretic peptide
CX3CL-1 ^1^	Corticotropin-releasing hormone (CRH) ^2^
IFN-γ ^1^	Chorionic gonadotropin
IL-1β ^1^	Dopamine (DA) ^2,3^
Il-2 ^1^	Endocannabinoids ^2^
IL-6 ^1^	Endorphins ^2^
MHC-I	Epinephrine ^2,3^
TLR2	Follicle-stimulant hormone (FSH)
TLR3	γ-aminobutyric acid (GABA) ^2^
TLR4	Glutamate ^2^
TNF-α ^1^	Growth hormone ^2^
	Insulin-like growth factor I (IGF-I) ^2^
	Luteinizing hormone-releasing hormone (LHRH)
	Luteinizing hormone (LH)
	Met-enkephalin ^2^
	Norepinephrine ^2,3^
	Oxytocin
	Prolactin ^2^
	Parathyroid hormone-related protein (PTHrP)
	Serotonin ^2,3^
	Substance P ^2^
	Thyroid-stimulating hormone (TSH)
	Vasoactive intestinal peptide (VIP) ^2^

^1^ The receptors for these ligands are expressed by neural cells, ^2^ the receptors for these ligands are expressed by leukocytes, ^3^ the membrane transporters of these ligands are expressed by leukocytes.

## Data Availability

Not applicable.

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
