# Peer review of "A “Drug-Dependent” Immune System Can Compromise Protection against Infection: The Relationships between Psychostimulants and HIV"

_viruses, 2021, doi:10.3390/v13050722_

Round 1

Reviewer 1 Report

The authors present a very broad range review of the relationships between psychostimulants and HIV. This is a valuable review describing how psychoactive substances influence the protection against HIV infection.

The topic of the manuscript is important. Undoubtedly, a review of the effects of drug use on the immune system provides important clues to HIV prevention strategies. There is however a need for more attention to the writing of the manuscript to connect each section. Little is mentioned about how psychoactive substances influence the molecular, cellular, pathways, and network level’s mechanistic understanding. This manuscript may need to employ a robust set of system biology information on data underlying transcriptional, epigenetic, and post-translational responses to explore the relationships between psychostimulants and HIV infection immunity.

The authors tried to touch on many aspects of the influence of psychoactive substances. The authors should discuss the relevance of specific effects of psychostimulants in innate and adaptive immunity as well. The psychostimulants of abuse activate common downstream sequences of events that have broad effects on cytokine and chemokine release through suppression of NF-κB activity and cause the loss of immune cells through apoptosis

Despite its depth, enthusiasm for the manuscript for the Viruses is tepid. A principal limitation is a degree to which it advances a mechanistic understanding of the immune response to psychostimulant use and the conclusions are quite broad and vague. 

Minor comment:

It might be a technical glitch. There are many unfit symbols shown in the table and text. (e.g CD3?, IL-1?TNF-? and IFN-? )

Author Response

We are grateful to referee for recognizing the value of our review and we have tried to address the deficiencies indicated in order to further improve it (e.g., attempting to better link each section). As the referee indicates, the review touches on many aspects of the influence of psychoactive substances and while we recognize that the recommendations made would help improve and extend the scope of the review (e.g., at the molecular and network level), we feel that the manuscript is already quite long and dense, and that delving deeper into these areas would perhaps be too much at this stage. We feel that these issues would be better dealt with in a follow-up article based on the foundations of this initial review, rather than overloading the reader at this point in time when we are presenting a summary of this relatively novel concept. As such, it would perhaps be better to address the influence of psychoactive substances at a systems biology level, as well as their transcriptomic, epigenetic and post-translational effects in a separate article focusing on more mechanistic details, which in turn could reach more focused conclusions. Indeed, the second reviewer requested that the review should be condensed rather than extended.

Regarding the issues with the symbols, this is indeed and error introduced by the processing of the manuscript files by the journal.

Reviewer 2 Report

In their manuscript, Amparo-Assis and coworkers discuss the links between psychostimulants, immune-associated cells and molecules and HIV infection. While the originality of the approach of these links by the authors is much appreciated, I feel that manuscript should be improved by shortening it at least by one third and focusing more explicitly on certain topics.

Major comments

1) Table 1: Please rename the column headings as the molecules mentioned can neither be identified as immune nor as neurotransmitters/modulator. Rather they have been classically associated with the immune and nervous systems, respectively.

2) Introduction, page 4, lines 140-143: “Indeed, certain neuropathologies and neuropsychiatric disorders have more recently been related to immune dysfunctions, and some of these disorders have been reversed by bone marrow transplantation [54,55], whereas in cases like depressive disorders an immunopathogenic cause has been proposed [56,57].” This is not that recent and has been proposed much earlier than the cited references suggest. Please, honor the first original studies indicating such links. 

3) Introduction, page 4, lines 170-172: “Other relevant molecules that have been studied at central levels are the toll-like receptors (TLRs), considered classical innate immune receptors that were recently implicated in certain drug actions on microglia.” Please, provide a reference for this important statement.

4) Catecholamines, page 5, line 189: “Catecholamines not only act as neurotransmitters but also, as immune transmitters” In my understanding catecholamines are neuromodulators rather than classic neurotransmitters. Please clarify.

5) Catecholamines, page 5, lines 206-207: “Both NE and DA meet the criteria to be considered transmitters in the immune system, although how immune cells regulate catecholamine synthesis is distinct.” Please provide details on what those criteria are.

6) Page 7, second paragraph. Please, discuss the recently-shown effects of VTA stimulation effects on immune system.

7) Page 10, line 469: “TLRs are not only immune system but also CNS receptors” This seems to suggest an opposition and may reinforce the wrong idea that there would be no immune system in brain.

8) Page 10, lines 479-481: “However, endogenous ligands have also been described for these receptors, and TLR4 and TLR2 can sense endogenous agonists such as heat-shock proteins and high-mobility group box-1 protein [160,161].” Would it be worthwhile to discuss the idea that fatty acids would be TLR ligands?

9) Page 16, lines 794-815: There seems to some contradiction between the start and the end of the paragraph. Please clarify

10) Part 4 should be dropped and just be mentioned in one or two sentences to set the scene in the introduction.   

Minor comments

11) The alpha symbol does not seem to have been generated corrected in the pdf file.

12) The abbreviation cART is somewhat confusing given the existence of CART Cocaine and Amphetamine-Related Transcript. Overall there are many abbreviations that are hard to keep track off. Please revise this.

Author Response

We would like to thank the reviewer for the thorough revision and the comments regarding our manuscript, as well as for all the suggestions that we feel have improved the presentation of the data.

In their manuscript, Amparo-Assis and coworkers discuss the links between psychostimulants, immune-associated cells and molecules and HIV infection. While the originality of the approach of these links by the authors is much appreciated, I feel that manuscript should be improved by shortening it at least by one third and focusing more explicitly on certain topics.

Thank you very much for these comments, we appreciate your recognition of our original approach to this topic, bringing together concepts from the fields of Neuroscience, Immunology and Virology. Since the audience of this journal is mainly comprised of virologists and infectologists, we considered it important to describe some basic aspects of Neuroscience and Immunology that may not be familiar to this audience, which is the main reason for the length of the manuscript. However, we have taken into account your advice and we have tried to shorten the manuscript by focusing on the main topics.

Major comments

1) Table 1: Please rename the column headings as the molecules mentioned can neither be identified as immune nor as neurotransmitters/modulator. Rather they have been classically associated with the immune and nervous systems, respectively.

Thank you for your advice. We have now modified Table I Legend as well as the column headings now read:

Table 1. The most relevant molecules classically associated with the immune system expressed by neural cells and classically associated with CNS expressed by immune cells

“Molecules classically associated with the immune system that are expressed by neural cells” (left column heading)

“Molecules classically associated with the CNS that are expressed by immune cells” (right column heading)

2) Introduction, page 4, lines 140-143: “Indeed, certain neuropathologies and neuropsychiatric disorders have more recently been related to immune dysfunctions, and some of these disorders have been reversed by bone marrow transplantation [54,55], whereas in cases like depressive disorders an immunopathogenic cause has been proposed [56,57].” This is not that recent and has been proposed much earlier than the cited references suggest. Please, honor the first original studies indicating such links. 

Thank you for pointing this out. To our knowledge, M Stein, M Irwin and AH Miller were pioneers in correlating immune system functionality and depressive symptoms. We had cited a revision from Irwin & Miller (2007) that provides a brief historical review of the initial works in this area. We have now added three further references related to their first relevant studies:

Ref # 60: Irwin, M.; Daniels, M.; Bloom, E.T.; Smith, T.L.; Weiner, H. Life events, depressive symptoms, and immune function. Am. J. Psychiatry 1987, 144, 437–441, doi:10.1176/ajp.144.4.437

.Ref # 61: Stein, M.; Miller, A.H.; Trestman, R.L. Depression, the immune system, and health and illness: Findings in search of meaning. Arch. Gen. Psychiatry 1991, 48, 171–177, doi:10.1001/archpsyc.1991.01810260079012..

.Ref # 62: Schleifer, S.J.; Keller, S.E.; Siris, S.G.; Davis, K.L.; Stein, M. Depression and Immunity: Lymphocyte Function in Ambulatory Depressed Patients, Hospitalized Schizophrenic Patients, and Patients Hospitalized for Herniorrhaphy. Arch. Gen. Psychiatry 1985, 42, 129–133, doi:10.1001/archpsyc.1985.01790250023003.

With regards bone marrow transplantation influencing psychiatric or neurological disorders, we have now added the three oldest references that we found related to this phenomenon:

Ref # 55: Lal, H.; Bennett, M.; Bennett, D.; Forster, M.J.; Nandy, K. Learning deficits occur in young mice following transfer of immunity from senescent mice. Life Sci. 1986, 39, 507–512, doi:10.1016/0024-3205(86)90506-0.

Ref # 56: Walkley, S.U.; Thrall, M.A.; Dobrenis, K.; Huang, M.; March, P.A.; Siegel, D.A.; Wurzelmann, S. Bone marrow transplantation corrects the enzyme defect in neurons of the central nervous system in a lysosomal storage disease. Proc. Natl. Acad. Sci. U. S. A. 1994, 91, 2970–2974, doi:10.1073/pnas.91.8.2970.

Ref # 57: Lescaudron, L.; Unni, D.; Dunbar, G.L. Autologous adult bone marrow stem cell transplantation in an animal model of huntington’s disease: Behavioral and morphological outcomes. Int. J. Neurosci. 2003, 113, 945–956, doi:10.1080/00207450390207759.

As a result, we have now no longer refer to these findings as “recent”, indicating that bone marrow transplantation can “modulate” rather than “reverse” psychiatric disorders, given that Ref # 55 provides evidence of the induction of learning deficits after the transfer of aged bone marrow (page 4, lines 133-136).

“Indeed, certain neuropathologies and neuropsychiatric disorders have been related to immune dysfunctions, and some of these disorders have been modulated by bone marrow transplantation (53-57), whereas in cases like depressive disorders an immunopathogenic cause has been proposed (58-62).”

3) Introduction, page 4, lines 170-172: “Other relevant molecules that have been studied at central levels are the toll-like receptors (TLRs), considered classical innate immune receptors that were recently implicated in certain drug actions on microglia.” Please, provide a reference for this important statement.

Following your comment, we now cite two references to support this statement on page 4, line 158:

Ref # 77 (that was originally cited in page 11, line 497):

Crews, F.T.; Zou, J.; Qin, L. Induction of innate immune genes in brain create the neurobiology of addiction. Brain. Behav. Immun. 2011, 25 Suppl 1, S4–S12, doi:10.1016/j.bbi.2011.03.003.

And Ref # 78 (that has been added now):

Crews, F.T.; Walter, T.J.; Coleman, L.G.; Vetreno, R.P. Toll-like receptor signaling and stages of addiction. Psychopharmacology (Berl). 2017, 234, 1483–1498

4) Catecholamines, page 5, line 189: “Catecholamines not only act as neurotransmitters but also, as immune transmitters” In my understanding catecholamines are neuromodulators rather than classic neurotransmitters. Please clarify.

            Thank you for raising this issue. We have revised our main bibliography regarding Neuropharmacology and catecholamines are considered as neurotransmitters. In the book “Molecular Neuropharmacology: A foundation for clinical Neuroscience” by Nestler, Hyman and Malenka (Nestler et al., 2009), a comparative figure is presented in Part 2, Chapter 7 (Figure 7-5) titled: “Comparison between classic neurotransmitter and neuropeptide system”, where they use dopamine as an example of a classic neurotransmitter.

We transcribe here the figure legend:

“Fig 7-5. Comparison between classic neurotransmitter and neuropeptide systems. Dopamine is used here to represent a classic neurotransmitter. A principal difference between these two systems is the cellular location of their synthesis. Although some dopamine is synthesized in the cell body, most is produced in nerve terminals. In contrast, neuropeptide synthesis begins in the cell body and continues as it is transported down the axon. Unlike dopamine, which is stored in small clear synaptic vesicles, neuropeptides are stored in large dense core vesicles. Both dopamine and the neuropeptide are enzymatically degraded, but only dopamine is transported back into the nerve terminal, where it is repackaged for subsequent release.”

            In addition, neuromodulators (mainly neuropeptides) are co-released with neurotransmitters, as can also be seen in Table 7-3 in the aforementioned book.

Similar concepts are described in Chapter 14 of Goodman and Gilman’s book “The pharmacological basis of therapeutics” (Brunton et al., 2019), where there is a detailed description of all the molecules considered to be neurotransmitters. In the description of neuropeptides (page 258), as specific distinction is made for modulators as they do not provoke direct excitation or inhibition but rather, a modulatory effect of the neurotransmitter co-released at the same synapsis. Neuropeptides and neurotransmitters are also distinguished on the basis of their mechanisms of synthesis, release and uptake, with neurotransmitter synthesized, stored and taken-up by the terminal, while the neuropeptides are synthesized in the soma, stored in dense core vesicles and degraded immediately after the release.

These criteria are also applied and catecholamines considered as neurotransmitters in a third book we consulted: “Psychopharmacology: Drugs, the brain and the behavior” (Meyer and Quenzer, 2005).

            As such, we consider it correct to describe catecholamines as classic neurotransmitters.

5) Catecholamines, page 5, lines 206-207: “Both NE and DA meet the criteria to be considered transmitters in the immune system, although how immune cells regulate catecholamine synthesis is distinct.” Please provide details on what those criteria are.

            In the light of your comment, we have added a sentence to page 5 (lines 197-202) that briefly describes the most important criteria used to verify a chemical’s status as a transmitter, as described by Meyer & Quenzer (see page 65: Meyer and Quenzer, 2005).

 “Both norepinephrine and DA meet the criteria to be considered as transmitters in the immune system, since the immune cells that contain them are capable of producing and inactivating them. Moreover, these cells release these transmitters upon stimulation, and there are receptors on both the target and releasing cells that are sensitives to agonists and antagonists [86]. However, how immune cells regulate catecholamine synthesis is different to that observed in neural cells.”

6) Page 7, second paragraph. Please, discuss the recently-shown effects of VTA stimulation effects on immune system.

Following your advice, we have now briefly discussed this topic on page 7 (lines 300-315).

“Data provided by 6-OHDA lesions in the NAc, that affect ventral tegmental area neurons through retrograde transport, demonstrated the participation of this central pathway in the immunomodulatory effects of amphetamine (65). The loss of dopaminergic terminals on NAc neurons not only reverses the peripheral effects of amphetamine on the lymphoproliferative splenic response but also, it blocks the increase in met-ENK levels in spleen and prefrontal cortex. Since the lesions involved affect the ventral tegmental area neurons, these dopaminergic neurons may also project to other brain areas, such as the prefrontal cortex and amygdala. In turn, these areas send GABAergic and glutamatergic projections to the ventral tegmental area (124). The pathway transmitting the messages evoked by amphetamine from the mesolimbic system to the immune system was not assessed in this study, although a possible link between the NAc and the extended amygdala in emotional control has been proposed. The modulation exerted by the amygdala (and hence by the NAc) on the SNS could reach the spleen and adrenal glands (122,123), and may therefore explain this phenomenon. This central modulation of the spleen through the SNS was also recently demonstrated via the activity of B-cells in a T-dependent immune response (98).”

7) Page 10, line 469: “TLRs are not only immune system but also CNS receptors” This seems to suggest an opposition and may reinforce the wrong idea that there would be no immune system in brain.

This is a very interesting point, thank you for mentioning it. Considering that TLRs can be expressed by neurons as well as being expressed by microglia in the CNS, clearly these receptors are not exclusive to the immune system even though they were initially described in immune cells. Thus, these TLRs indicate that there are receptors shared by both systems that can help “detect internal situations” to which the cells must react accordingly. These responses go beyond the classic roles of these receptors in “pathogens defense”, as defined by the word “immune”. Please note that this idea was mentioned in the last paragraph of this section (see lines 474-489). To highlight this concept and in the light of your suggestion, we have now changed the subheading on page 10 (line 458):

“2.3.1. TLRs as immune and CNS receptors”

We have also changed the subheadings related to catecholamines and enkephalins on page 5 (line 181) and page 8 (line 348), respectively:

“2.1.1. Catecholamines as neuro-and immune-transmitters”

“2.2.1. Enkephalins as neuro-and immune-modulators”

8) Page 10, lines 479-481: “However, endogenous ligands have also been described for these receptors, and TLR4 and TLR2 can sense endogenous agonists such as heat-shock proteins and high-mobility group box-1 protein [160,161].” Would it be worthwhile to discuss the idea that fatty acids would be TLR ligands?

Following your advice, we have mentioned that saturated fatty acids act as TLR ligands on page 10 (lines 470-471):

“However, endogenous ligands have also been described for these receptors, and TLR4 and TLR2 can sense endogenous agonists like heat-shock proteins and high-mobility group box-1 protein (174-175). More recently, saturated fatty acids have also been described to be TLR4 ligands (176).”

and we refer to a detailed review of this topic, such as Ref #176:

“Rocha, D.M.; Caldas, A.P.; Oliveira, L.L.; Bressan, J.; Hermsdorff, H.H. Saturated fatty acids trigger TLR4-mediated inflammatory response. Atherosclerosis 2016, 244, 211–215.”

9) Page 16, lines 794-815: There seems to some contradiction between the start and the end of the paragraph. Please clarify.

We thank you for pointing this out and as such, we now describe the data more comprehensively on page 17 (lines 784-815):

“Studies regarding TLR4 activation by bacterial infection in people living with HIV may shed light on how a TLR4 agonist can influence HIV evolution. Early in vitro studies showed that a purified protein derived from Mycobacterium tuberculosis increased viral mRNA expression in HIV infected monocytes, although LPS failed to induce the detection of HIV RNA in these primary monocytes from 24 h to 5 days (256). Bacterial LPS has been shown to activate TLR4 and induce HIV-long-terminal repeat (LTR) transactivation (257), although other TLRs may also mediate this effect (258,259). However, later studies demonstrated that repeated LPS exposure inhibits HIV replication, as witnessed by weaker expression and protein production of HIV p24, concluding that TLR4 ligands can induce tolerance and suppress HIV-LTR transactivation in human monocytic cell lines (260). Along similar lines, the participation of TLR4 in HIV infection of macrophages in the genitourinary mucosa was recently reported, playing a key role in HIV sexual transmission and pathogenesis (261). Since Neisseria gonorrhoeae augments mucosal transmission of HIV, both by inducing inflammation, and by directly activating virus infection and replication, the interaction between commensal (E. coli) or pathogenic (N. gonorrhoeae) bacteria on HIV-latency has been studied in macrophages. Both these bacteria encode TLR2 and TLR4 ligands, and they repress HIV replication in macrophages, inducing a latency-like viral state. Soon after TLR4 stimulation, cellular mechanisms involving IRF1, NF-kB and HIV Tat lead to high levels of virus replication, whereas at later time points IRF1/IRF8 heterodimers repress HIV replication. Thus, productive HIV infection of macrophages seems to be altered by TLR signaling, with HIV activated by TLR2 and TLR5 signaling, and HIV replication repressed by TLR3 and TLR4. These data suggest that TLR4-mediated signaling (in this case induced by a microbe) represses HIV replication and induces viral latency in human macrophages, contributing to the establishment and maintenance of latent HIV infection in these cells. As such, these macrophages contribute to the viral reservoir in people living with HIV under ART (261). This interesting data led us to suggest that chronic administration of other TLR4 agonists (e.g., cocaine or amphetamine) might also induce a similar effect on the TLR4 expressed by macrophages, microglia or even CD4+ T cells. Such a phenomenon would have certain clinical implications for the establishment and maintenance of the latent HIV reservoir, both in the periphery and the CNS. Furthermore, the participation of TLR2 should also be considered in this process as cocaine enhances its expression by microglia (197).”

10) Part 4 should be dropped and just be mentioned in one or two sentences to set the scene in the introduction.

Thank you very much for your observation. Following your request, we have removed Part 4 and added the following information to the Introduction (page 4, lines 158-166).

“On the other hand, HIV infection can aggravate the impact of the rewarding effects of psychostimulants since the HIV Tat protein produces a direct but reversible inhibition of DA transporter (DAT) activity in rat striatal synaptosomes (79,80). Tat expression can provoke behavioral cross-sensitization to the locomotor effects of methamphetamine (81), and the long-term impact of Tat on the DA transmission and its drug reinforcing effects may impair reward function, helping sustain the drug use/abuse that can lead to addiction (82). However, all these findings were derived from the CNS and little is known about Tat’s effects at peripheral sites where DAT is expressed, such as immune cells.”

Minor comments

11) The alpha symbol does not seem to have been generated corrected in the pdf file.

We assume that this problem was produced by the change in format during the journal’s handling of the files submitted. We have now corrected this and we hope the editors will take this into consideration in the future.

12) The abbreviation cART is somewhat confusing given the existence of CART Cocaine and Amphetamine-Related Transcript. Overall there are many abbreviations that are hard to keep track off. Please revise this.

            In view of your comment we have replaced cART with ART since both abbreviations are widely used in HIV bibliography. We have also removed abbreviations for: people who inject drugs (PWIDs); men who have sex with men (MSM); people living with HIV (PLWH); adrenoreceptors (ARs); norepinephrine (NE); tyrosine hydroxylase (TH); prefrontal cortex (PfC); ventral tegmental area (VTA); long-term potentiation (LTP); opioid growth factor (OGF); dendritic cells (DCs); antigen-presenting cells (APCs); pattern-recognition receptors (PRRs); pathogens-associated molecular patterns (PAMPs); danger-associated molecular patterns (DAMPs); elite controllers (ECs); and post-treatment controllers (PTCs).

References:

Brunton, L., Knollmann, B., Hilal-Dandan, R., 2019. Goodman and Gilman’s the Pharmacological Basis of Therapeutics, 13th ed. McGrawHill.

Meyer, J.S., Quenzer, L.F., 2005. Psychopharmacology. Drugs, the brain and the behavior. Sinauer Associates, Inc.

Nestler, E., Hyman, S., Malenka, R., 2009. Molecular Neuropharmacology: A Foundation for Clinical Neuroscience, Second. ed. McGrawHill.

Rocha, D.M., Caldas, A.P., Oliveira, L.L., Bressan, J., Hermsdorff, H.H., 2016. Saturated fatty acids trigger TLR4-mediated inflammatory response. Atherosclerosis. https://doi.org/10.1016/j.atherosclerosis.2015.11.015
